# Diversity of Plant Sterols Metabolism: The Impact on Human Health, Sport, and Accumulation of Contaminating Sterols

**DOI:** 10.3390/nu13051623

**Published:** 2021-05-12

**Authors:** Arthur T. Kopylov, Kristina A. Malsagova, Alexander A. Stepanov, Anna L. Kaysheva

**Affiliations:** Institute of Biomedical Chemistry, Group of Biobanking, 10 Pogodinskaya Str., Bld. 8, 119121 Moscow, Russia; f17-1086@yandex.ru (K.A.M.); aleks.a.stepanov@gmail.com (A.A.S.); kaysheva1@gmail.com (A.L.K.)

**Keywords:** sterols, biotransformation, cholesterol, food matrix, diet, anabolic effect

## Abstract

The way of plant sterols transformation and their benefits for humans is still a question under the massive continuing revision. In fact, there are no receptors for binding with sterols in mammalians. However, possible biotransformation to steroids that can be catalyzed by gastro-intestinal microflora, microbial cells in prebiotics or cytochromes system were repeatedly reported. Some products of sterols metabolization are capable to imitate resident human steroids and compete with them for the binding with corresponding receptors, thus affecting endocrine balance and entire physiology condition. There are also tremendous reports about the natural origination of mammalian steroid hormones in plants and corresponding receptors for their binding. Some investigations and reports warn about anabolic effect of sterols, however, there are many researchers who are reluctant to believe in and have strong opposing arguments. We encounter plant sterols everywhere: in food, in pharmacy, in cosmetics, but still know little about their diverse properties and, hence, their exact impact on our life. Most of our knowledge is limited to their cholesterol-lowering influence and protective effect against cardiovascular disease. However, the world of plant sterols is significantly wider if we consider the thousands of publications released over the past 10 years.

## 1. Introduction

Plant sterols are inherent compounds of many nutritional supplements and food additives. Sterols are chemical compounds based on 1,2-cyclopentaneperhydrophenantrene and are characterized by hydroxyl moiety at the 3C position and the side chain at the 17C position. The latter makes them structurally similar to pregnenolone, which is a fundamental molecule for all 17-ketosteroids generation.

Most studies about the supplement of phytosterols are devoted to their efficacy in lowering of low-density lipoprotein cholesterol (LDL-C) [1]. When discussing the role of phytosterols in diet, several fundamental aspects affecting their clinical efficiency are considered, including factors associated with the delivery of phytosterols, food matrix, dosage, frequency of intake, and chemical forms of these compounds, i.e., sterols or stanols [2]. The responsiveness to plant sterols and their bioavailability also depends on age and gender, and are determined by nutrigenetic differences in metabolic factors affecting the uptake of cholesterol, activity of cytochrome CYP7A1 and APOE gene expression [3].

The exact impact of sterols on our life seems controversial. A positive influence on cholesterol metabolism, lowering of triglycerides, and immune-modulating properties are well-known [3,4,5]. In this respect, plant sterols are readily recommended as adjustments to diet and as dietary agents that can lower risk of cardiovascular disease, render anti-atherogenic effect, preserve oxidative stress, and adjust or normalize endogenous cholesterol uptake. Experts in diet and nutrition agree that plant sterols are the most proper and abundantly occurring agents for balancing health and might be considered as a functional food supplement [2]. On the other side, there are plenty of reports about the possible adverse effect of sterols on women’s health and their transformation to anabolic-like human steroids. 

In overwhelming majority, the great variability in plant sterols effect is explained by differences of individual phenotypes and gut microbiota composition that participates in transformation of sterols into secondary metabolites, with potentially higher activity and impact on health [6,7,8]. Several studies, including randomized clinical examination, reported that people with a plant-based diet model have a lower risk of metabolic syndrome and cancer [9], although it comes at odds with recommendations of American Diabetic Association and relationship with lipids profile in patients with type 2 diabetes mellitus [10,11] as well as anti-cancer protective effect, which has been revealed on a limited number of studies [12].

Plant sterols draw the attention of researchers and the clinical community due to their high potency in therapy and a wide diversity of properties. Nevertheless, despite significantly raised knowledge about sterols for the past decade [13,14,15,16,17], the exact mechanisms of cholesterol-lowering, anabolic-like, and anti-atherogenic effects are still not well understood, but are known to be sensitive to cholesterol trafficking, activity of reducing/hydroxylating enzymes, MAPK and Akt-signaling pathways. Therefore, this review focuses on the variety of plant sterols’ impact on human health and the possible mechanisms of their action mentioned in different clinical trials, meta-analysis and research studies starting from the year 2010.

## 2. The Prevalence and Variety of Plant Sterols

The minimal intake of plant sterols per day should be at least 1 g to produce an appreciable cholesterol-lowering effect [18]. However, fruits, vegetables, and plant oils are hard to be considered as natural sources of plant sterols due to the insufficient content of these compounds. For example, the content of naturally occurring sterols in fruits and vegetables is ranged between 38 to 439 mg/kg of fresh weight, whereas in grains, their content reaches up to 1780 mg/kg [18,19]. Thus, about 2 kg of fruits and vegetables or about 1 kg of grains has to be consumed daily to obtain the necessary 1 g of plant sterols. Other nutrition experts argue that 200 to 400 mg of plant sterols per day is the optimal dose for handling diet and being processed by the gastrointestinal tract [20]. A typical western diet contains about 300 mg of plant sterols in daily intake [21] and, accordingly, other sources including corn fiber oil, soy, rapeseeds, or rice bran oil are considered as the alternative feedstock of plant sterols because they contain up to 15% of plant sterols [18,22].

About 300 various species of plant sterols are described [23,24,25], which differed by side chain at C24 position and can be specific for certain plant species. This variety of compounds covers free sterols and their ester conjugates with fatty acids at the 3β-OH moiety, acylated sterol glycosides, and sterol glycosides (typically with glucose), which are specifically absent in all animals. The most abundant and typical plant sterols are represented by campesterol and stigmasterol. The main plant enzymes responsible for maintaining the balance of different sterols and their biosynthesis are 3-hydroxy-3-methylglutaryl-CoA reductase, C24-sterol methyltransferase, and C22-sterol desaturase. A human can discriminate non-cholesterol plant sterols and excrete most of them with feces together with bile acids, which suggests the existence of a mechanism responsible for retaining endogenous cholesterol over plant sterols [26].

## 3. Factor Associated with the Efficacy of Plant Sterols Absorption

The clinical efficacy of plant sterols is predominantly associated with their cholesterol-lowering effect. Numerous recent studies were dedicated to important aspects (chemical form of sterols and supplements, food matrix, way of delivery, and frequency of intake) influencing sterols absorption and their action efficacy [27,28,29].

No significant difference in plant sterol efficacy was established when supplemented with low-fat or high-fat foods [30] as wells as no difference being found between the consumption of plant sterols or stanols. The cholesterol-lowering effect for both sterols and stanols was estimated as 0.34 mmol/L if the consumed mean daily dose was 2.15 g. However, an apparent tendency (*p* = 0.054) of lower efficacy was established for single vice multiple intakes of plant sterol supplements. Other research groups found that the intake of different dietary plant sterols leads to a comparable decrease in plasma cholesterol, but, in contrast, stanols are up to 50-fold less effective [31,32].

The use of food supplements with plant sterols together with the lipid-lowering therapy demonstrated an additive effect. When atorvastatin and plant sterols were used combined, the lowering effect of total cholesterol and LDL-C was estimated at 22% and 38%, respectively (*p* < 0.05). In contrast, the effect of monotherapy by the lipid-lowering agent in a dose of 40 mg was 3% and 22% for total cholesterol and LDL-C, respectively. Notwithstanding, some epidemiological studies reported the increased risk of cardiovascular events and the net negative effect if the plasma level of plant sterols reaches the upper normal [24,33].

A meta-analysis of a large number of studies (*n* = 124 studies in total) established the difference in plant sterols absorption efficacy depending on the type of food matrix [34]. The effect of plant sterols and their absorption was more pronounced when consumed with drinking food (9.5% lowering of cholesterol, *p* < 0.001) rather than with solid food. Moreover, the effect was more explicit when sterols are consumed at lunch instead of breakfast. Although it seems on the surface that the high-fat food matrix is the most appropriate carrier for sterols absorption, low-fat may also be competent if products contain emulsifiers [35]. Therefore, the total cholesterol-lowering effect and rate of sterols absorption were irrespective of total fat content in the meal [36]. Nevertheless, a robust dose-dependent response was observed when sterols and stanols were analyzed separately. No significant difference was detected between the efficacy of free and esterified sterols (less than 2% in a dose between 0.6 and 3.3 g per day), thus plant sterols and non-esterified stanols contribute almost equally in cholesterol-lowering effect [31,37].

Since many food supplements and additives with plant sterols are currently manufactured in the form of capsules or pills, this raises the question about their effectiveness compared to the regular intake. The meta-analysis carried in the 2013 year and covered eight eligible clinical trials showed a similar effect consisting of 0.31 mmol/L (*p* < 0.0001) cholesterol-lowering for both dietary sterols and sterols delivered in capsules or pills [38,39]. However, it should be warned that the majority of studies lack the information about particle size, solubility, and bioavailability, which are all critical characteristics when discussing capsule and pill formulation as possible carrier agents [40,41,42]. Different studies indicated that an excretion rate of sterols and cholesterol is the most critical factor affecting intestinal absorption of plant sterols [43,44,45]. The excretion rate is determined by ATP-binding cassette sub-family G member 5/8 transporter (ABCG5/ABCG8) and only brain cells are the exclusion of this rule (Figure 1).

Liver X receptor (LXR) is the primary positive regulator of these transporters, and mutations in either of these genes cause the condition of sitosterolemia, characterized by the accumulation of cholesterol and plant sterols in plasma and tissues [46]. It has been found that only up to 2% of plant sterols enter the system, whereas the rest is pumped into the bile [47]. At least in animal models with the deleted *Abcg5* and *Abcg8* the increase of dietary plant sterols absorption in 2–3-fold changes and associated 30-folds increase in plasma cholesterol has been demonstrated [48]. Alternatively, it was reported that murine deficiency in ABCG5/ABCG8 demonstrated the level of plasma and liver sterols comparable with the wild-type mice; however, the level of non-cholesterol sterols was elevated up to 30-folds, indicating that sterols are preferentially secreted into bile. Besides, some authors revealed that such a population of knockout animals is characterized by the decreased mRNAs level for 13 enzymes involved in cholesterol biosynthesis [48,49].

Assumingly, disruption of even the *Abcg5* gene alone is already enough to increase meaningfully the rate of dietary sterols absorption, while maintaining the rate of cholesterol secretion steady. The genome-wide associated study provided by three European biobanks (Iceland, Denmark, and the UK) confirmed that nine rare polymorphisms in *Abcg5/8* loci are tightly associated with the rate of plant sterols absorption, their excretion rate, level of LDL, and the risk of coronary artery disease [50].

The affinity for and the solubility in bile salt micelles born completely different opinion regarding the factors affecting absorption of plant sterols. The suggestion came from the upshot of experiments on human colorectal adenocarcinoma cells (Caco-2), when intestinal absorption of cholesterol was inhibited because of plant sterols being well-solubilized in bile salts and, thence, they competitively limit the solubilizing capacity for cholesterol [51].

## 4. Benefits of Plant Sterols in Clinical Application

A critically important issue is the controversial information regarding the benefits and risks of regular intake of plant sterols for the health. Generally, it takes a balanced position with multifaceted aspects.

In 1954, the very first publication declared the positive sustained cholesterol-lowering effect of β-sitosterol in nine subjects under the unrestricted diet. Authors immediately proposed the interference of cholesterol with the absorption of dietary plant sterols, and reported the lack of any toxic effect after the sterols-enriched diet [52]. Since that, almost 1500 publications about the cholesterol-lowering effect and more than 4000 publications about other distinct effects of plant sterols have arrived in the PubMed library. There are many research groups, who claim about the favorable [53,54] or unfavorable [55,56,57,58] actions of plant sterols on arterial function, atherosclerosis and risk of cardiovascular disease (CVD) [59,60], immune stimulation [61,62], cutaneous wound healing [63,64] and the impact on the central nervous system [65,66,67] (Table 1).

### 4.1. Cardioprotective Property of Plants Sterols

Apparently, triglyceride- and cholesterol-lowering effects are the most examined amongst other clinical concerns. Therefore, we make just a short stop on this aspect because a plethora of publications have already discussed exactly this matter [25,33,38,39,52,53,54,59,60]. Some clinical trials indicated that a daily intake of dietary plants sterols at a dose of about 2.5 g per day is sufficient to decline the level of circulating triglycerides by 9.5% [69], and low-fat soy beverage may assist in 13% lowering of LDL-C [90]. The meta-analysis of 124 studies revealed that the average intake dose of cholesterols in the population is about 0.2 to 9.0 g per day, whereas a dose of 0.6 to 3.3 g per day is already sufficient to reduce the serum level of LDL-cholesterol [34], and the effect continues to increase up to 12% with a total intake of 3 g per a day of sterols or stanols. It suggests a dose-dependent cholesterol-lowering effect of dietary sterols. Such dose-dependent efficiency was observed in comparative study with human subjects, who were sub-grouped according to the main sterol-enriched product chosen for the supplementation for 4 weeks [91,92]. It has been demonstrated that the serum level of cholesterol gradually decreased if consuming 3, 6, or 9 g of sterols and stanols-enriched food, and achieved up to 17.4% lowering effect at a dose of 9 g of sterols per a day.

Therefore, the elaboration of a bespoke therapy strategy combining dietary plant sterols with statins, which inhibit cholesterol synthesis, is actually of growing interest [93,94]. Notwithstanding, due care must be taken if accepting hypolipidemic therapy in combination with sterols, since statins, on the one hand, inhibit the synthesis of cholesterol but, oddly, increase the absorption of cholesterol. Consequently, plant sterols may increase the chance of undesirable cardiovascular events in subjects with a high cholesterol level.

Assumingly, a wide range of the efficient sterols-dose tracked in numerous research studies and clinical trials (Table 1) proposes the existence of great interindividual variability in responsiveness to dietary sterols. A feeding controlled clinical trial for healthy males and female aged between 19 and 60 years old, who received 1% soymilk and low-fat soy beverage dairy for 4 weeks, exhibited LDL-C lowering responsiveness ranging between −33% and +38% [90]. As has been touched on before, the response variability can be caused by the food matrix and some other food-intake factors (formulation, daily consumption, frequency, etc.). However, other metabolic-related and nutrigenomic factors may also determine the plants sterols absorption and cardioprotective efficiency action. The most obvious reason is tightly linked with polymorphisms of genes involved in cholesterols transport, tissue trafficking and transformation, including APOE, Niemann-Pick C1-like 1 (NPC1L1), cytochromes P450 and ABCG5/G8 transporters [3,95], but no gender-specific difference in responsiveness was determined after review of several clinical trial studies [1]. Liver cytochromes and enzymes make a significant input in availability of plant sterols, since some of them can be absorbed and transported as their native form. Nevertheless, the majority of ingested plant sterols are esterified or polymerized, or have a glycosyl moieties, and thereby must be carried by hydrolyzing enzymes or gut microbiota to be captured for transport and transformation [96]. There is also another challenge greatly contributing in reported interindividual variability of cholesterol-lowering effect, and related to methodological variability of measurement of cholesterol metabolism using plant sterol concentration, which makes it difficult to compare the measured values between different laboratories [93].

Hence, before recommending or prescribing cardioprotective food-based therapy, multiple factors, including pharmacogenomic, have to be taken into account. At the same time, so far, the issue of response variability is not completely resolved in scientific-based way, and the food industry has a great potency in the development of pre-processing and optimized functional food with a higher bioavailability regardless of the personal responsiveness of the consumer.

### 4.2. Neuroprotection and Neuroimmunomodulation

The impact of plant sterols on the central nervous system is another exciting point. The good news is that there is no adverse effect on mental or cognitive activity; it is either positive or vague. It happens mostly because of two reasons. First, plant sterols are poorly transported across the blood-brain barrier (BBB); thus, their effects on the brain and neurons are most likely weak or limited [71]. Second, the mechanism of plant sterols actions on brain cells are mostly unclear; hence, we still may not surmise about possible mediated adverse effects.

The mechanism by which plant sterols cross the blood-brain barrier and enter the brain cells remains mostly a hypothesis. The ABCG5/8-mediated transport is almost excluded, because these genes are not expressed in brain cells and in the BBB environment (Figure 2). Animal models suggested that the most credible way of entering is associated with the uptake of cholesterol via high-density lipoprotein cholesterol (HDL-C) receptors, which are abundantly expressed on basolateral compartments of the BBB, and involves translocation mediated by apolipoprotein A1 and ATP-binding cassette transporter A1 (APOA1/ABCA) [97]. At the same time, animals deficient in ABCG5/8 transporters also demonstrated elevated (up to 16-fold) levels of plant sterols both in serum and in brain cells [73]. However, it should be emphasized that the metabolism of cholesterol in the brain is broadly different from those in the periphery and accounts for no more than 1% of the peripheral turnover due to restrictions of BBB [97,98]. The requirement of brain cells in cholesterol is covered by de novo synthesis while the excess is removed through the conversion to the polar 24(S)OH-cholesterol and its secretion via the BBB as the main pathway of cholesterol maintenance in the brain cells [99].

Unlike the endogenous cholesterol, plant sterols are not transformed into more polarized derivatives due to the alkyl moiety allocated at the C24 position. Therefore, after entering the brain circulation, plant sterols irreversibly accumulate and incorporate into cell membranes [80]. It is known that 24(S)OH-cholesterol downregulates de novo synthesis of cholesterol in a post-transcriptional manner through SREB2, but upregulates synthesis of apolipoprotein E through stimulating of LXRα and LXRγ receptors (*vide supra*) [75]. Evidence exists that APOE deficient animals display a significantly increased serum level of plant sterols accommodated in very-low-density lipoprotein (VLDL) particles, but in the brain the level of dietary sterols remains almost unaltered since they cross BBB in HDL-C complexes (Figure 1 and Figure 2). However, gradually increasing the concentration of circulating sitosterol affects BBB in a time-dependent manner and, thus, producing a sterol type-specific effect in brain cells [73]. The obvious role of APOE in the transport of dietary sterols has been demonstrated in homozygous ApoE^−/−^ mice (Table 1). Although the subjected substance was 7βOH-sitosterol, which is a product of sitosterol transformation, its concentration in the brain increased up to 65.8-folds (*p* < 0.001; Table 1) compared to wild-type mice [75]. Nevertheless, the authors concluded that despite the dramatically increased concentration of 7βOH-sitosterol, both the product and the initial substrate (sitosterol) do not influence the brain circulation of cholesterol.

The integrity of BBB is another essential aspect regulating plant sterols transport. Experiments on animals deficient in platelet-derived growth factor subunit B (PDFGB) showed a substantial enrichment of brain cells with [2H]-labeled dietary campesterol and sitosterol after 10 to 40 days of feeding due to the leakage of disrupted BBB [72]. Additionally, it has been observed that the complexity of the sterols side chain could determine the ability to cross BBB [100].

Admittedly, the exact role of dietary plant sterols is not yet recognized; thus, it cannot be judged as strictly positive or detrimental. Long-term exposure to plant sterols leads to an increased level of sterols in brain cells and sufficiently reduces the density of amyloid plaques in patients with Alzheimer’s disease but, *bona fides*, does not lead to the improvement of cognitive function and memory [81,101]. Likewise, mice fed with a stigmasterol-enriched diet exhibited auspicious cleavage of amyloids and reduced generation of amyloids (62.00% ± 3.9%, *p* = 0.0005) caused by the declined activity of β-secretase [102]. However, the effect was apparently sterols-specific because β-sitosterol increased the secretion of amyloid by 115.2% ± 2.2% (*p* < 0.001), whereas campesterol and brassicasterol had a neglected impact on the activity of β-secretase [102].

Since the early stage of Alzheimer’s disease is associated with the impairment of BBB functioning, the reduced concentration of plant sterols in cerebrospinal fluid (CSF) was supposed as a promising predictive biomarker with an 85% sensitivity and 75% specificity [103,104]. However, the cross-sectional clinical trial carried on patients with Alzheimer’s disease in Sweden demonstrated that brain levels of dietary sitosterol and campesterol were almost indistinguishable compared to the aligned control group of healthy donors (6.3 ± 0.8 ng/mg and 6.2 ± 0.8 ng/mg vice 5.0 ± 0.8 ng/mg and 5.0 ± 0.8 ng/mg, *p* < 0.05, in the assayed and the control groups, respectively). This suggests the integrity of BBB, but the level of 27OH-cholesterol, as the indicator of potential oxidative damaging, was dramatically increased in such patients [105].

Application of plant sterols for the management of neurodegenerative diseases therapy was proposed due to their immune-modulating activity engaged in the augmentation of bone-marrow-derived macrophages polarization to anti-inflammatory phenotype. However, results were contradictory because the effect of brassicasterol revealed a depletion of IL-10α (*p* < 0.01) but the sustained elevation of interleukins IL-1β, IL-6, and IL-12 in 16 days after treatment [106]. On the contrary, model animals with the autoimmune encephalomyelitis demonstrated a 10% increase of IL-10α (Table 1), but the macrophages entering and the level of other pro-inflammatory cytokines, including IL-6 and tumor necrosis factor-α (TNF-α), moderately decreased, thereby showing a suppressive effect of plant sterols on the immune activity in the central nervous system [107]. Yet, since neuroinflammation is a pivotal mediator of neurodegeneration and hinders neurons repair, one can suggest the positive suppressive effect of brassicasterol on a humoral immune response that may ameliorate selected symptoms of neurodegenerative diseases (Table 1).

Since both Alzheimer’s disease and multiple sclerosis (MS) are tightly associated with the lipid metabolism, recent findings demonstrate the ability of plant sterols to restore the impaired uptake of cholesterol. Despite the close relationship between these two pathologies, MS is more overtly characterized by demyelination, chronic inflammation, and the development of autoimmune response against CNS antigens. These symptoms in MS patients grow rapidly for a relatively short time. The disease is more prevalent among women than in men and affected about 2.5 million people worldwide in 2019 [108]. Turning to the panoply of evidence-based data that cholesterol is a limiting factor in remyelination, Sher G. et al., supposed that plant sterols additives may stimulate the myelination process and adapted the suggestion to an animal model [109].

Another study determined that dietary sterols subjected for two weeks at a dose of 2% (*w*/*w*) markedly improved myelination and density of oligodendrocytes in cuprizone animal model but barely changed the level of serum cholesterols (increased to 79 ± 3 mg/dL compared to 72 ± 6 mg/dL in the control group of animals). Results of RT-PCR assay suggested that the dietary sterols modulate a broad range of growth factors involved in the differentiation and survival of oligodendrocyte precursor cells. While *Fgf1* and *Shh* were elevated after supplementation with sterols, *Igf1*, *Egf* and *Cntf* were almost unaffected by sterols but regulated by cuprizone [82]. The apparent positive effect of plant sterols on the rate of myelination in MS models agrees with the finding of the decreased cholesterol synthesis after monotherapy treatment with statins inhibiting myelinations [110,111].

Another side of plant sterol anti-inflammatory property is faced with their ability to act as ligands for LXR and peroxisome proliferator-activated (PPAR) receptors. Such sterol-mediated stimulation positively regulates cholesterol metabolism due to the enhancement of APOE expression and the negative impact on the NF-kB (nuclear factor kappa) pathway that determines the expression of pro-inflammatory cytokines [112,113]. This interplay improves the uptake of cholesterol and encourages the process of myelination. In particular, sterols from *Aloe vera* boost the expression of *Fatp1*, *Acox1,* and *Cpt1* in a dose-dependent manner (Table 1) through the ligand-binding activity targeted toward PPAR receptors [83]. This activity produces a sufficient antioxidant response resulting in the increased level of glutathione and the diminished expression of IL-18 [84]. In summary, numerous investigations and reviews bear evidence about the positive influence of plant sterols on the recovery and amelioration of patients with neurodegenerative diseases. Assumingly, such an effect is accomplished via the normalization of cholesterol synthesis in oligodendrocytes or caused by the specific reduction of amyloid production, reflected in blunting of the inflammatory condition and contemporaneous activation of regenerative cascades.

### 4.3. Anti-Aging and Skin Regeneration Effects

Cholesterol maintenance and amelioration of neurodegenerative conditions are not the only applications of plant sterols. Skin regeneration and wound healing are becoming one of the largest fields, where dietary sterols are widely utilized and originated from traditional Thai and Chinese medicine. The biochemical investigation indicated that skin injury regeneration is accelerated through the anti-inflammatory properties of plant sterols in the same manner as has been touched above for neurodegenerative diseases. Ethanolic extract of traditional Thai medical plant parts (seeds, root, and pericarp) obtained from *Garcinia mangostana* L., *Glycyrrhiza glabra* L. and *Nigella sativa* L., exhibited an inhibiting property toward superoxide dismutase (SOD) and nitric oxide (NO) with IC50 equal to 71.54 ± 3.22 µg/mL and 78.48 ± 4.46 μg/mL, correspondingly. Moreover, extracts from *G. mangostana* and *G. glabra* at a subtoxic concentration of 5 µg/mL (Table 1) were capable of enhancing the proliferative activity of 3T3-CCL92 murine fibroblast cells on 52.68 ± 1.99% [114], which affirmed the expected positive effect of plant sterols extract on regeneration and recovery after damage. However, it should be admitted that the net positive impact can be caused by the complex influence of many compounds that were co-extracted with plant sterols in a crude preparation. Hence, the role of the exact plant sterols in the considered case is debatable.

A more detailed study has been accomplished on animals who were given a dermal wound and further treated on topical application with brassinosteroid. Dermal regeneration and proliferative activity were evaluated for 10 days by inspecting factors involved in the regulation of cell proliferation and cytokines release. As demonstrated, the mRNA level of TNF-α was almost as suppressed in the treated mice, whereas transforming growth factor-β (TGF-β) was significantly lower compared to the untreated animals [46]. Based on this evidence, authors supposed that the enhancement of proliferative activity and fibroblasts migration is accomplished via activation of the PI3K/Akt (phosphoinositide 3-kinases/protein kinase B) signaling pathway (Table 1).

Before, the selective activity of brassinosteroids toward PI3K/Akt signaling has been shown in similar experiments that purposed to promote growth factors during regeneration of skeletal muscle. Besides, it was established that some of the brassinosteroids can promote proteins synthesis more than on 37% (*p* < 0.001) and decrease proteins degradation concurrently, which is comparable with the influence of insulin growth factor-1 (IFG-1) at a concentration of 6.5 nmol/L and gives a total effectiveness of around 42% [85]. To answer what is the pharmacogenomic effect between the treated and the control animals, authors conducted a PCR array analysis from the gastrocnemius muscle biopsies. Expectedly, genes (*Adra1d*, *Igfbp1*, *Srebf1*, *Fbp2*, and *Igf2*) acting in positive regulation of mitogen-activated protein kinase (MAPK) and PI3K/Akt pathways, and several myogenic transcriptional factors, were found as the most significantly upregulated during muscle cells proliferation. However, the exact targets of brassinosteroid that trigger downstream signaling are still uncertain.

Yet, the observed and repeatedly demonstrated ability of plant sterols to enhance the rate of protein synthesis and to diminish the rate of protein degradation aspires to persuade us of the potential anabolic effect of dietary plant sterols. Furthermore, it is well known that activation of Akt signaling may sufficiently support the increase of muscle mass (up to 50% within three weeks on the targeted stimulation) with no obligation to activate satellite cells [115].

## 5. Dietary Sterols in Sport

The potential anabolic properties of plants are very attractive in professional sport. Considering the growing offers of plant-based food supplements and additives intended to the intentional improvement of elite athlete abilities, WADA (World Anti-Doping Agency) included some suspected dietary sterols in the monitoring list. Among them, ecdysterone (20β-OH-ecdysone) takes a specific role, because in 2019 WADA reported that ecdysterone may act as a non-conventional anabolic steroid and its properties are mediated through the binding with estrogen receptors, thus, revealing to be even more effective in comparison to some synthetic steroids. It has been reported that the study of ecdysterone supplementation resulted in the increased level of IGF-1 (as for brassinosteroids) but declines the serum level of thyroxine (T4). Therefore, since 2008, WADA is prone to include ecdysterone in section S1.2 “Other anabolic steroids” of the WADA’s prohibited list, and its administration is fraught with a risk of abuse [116]. This assertion remains highly speculative because targeted studies of ecdysterone pharmacokinetics and pharmacodynamics on humans are sparsely represented. However, the suspicion about possibly adverse properties of dietary sterols has a background.

There is an accumulating number of reports about the targeted effect of 20β-OH-ecdysone on muscle cells size and elevation of a myonuclear number of the regenerating myocytes, which are very convincing in anabolic and modulating properties of this sterol, albeit mediated in an unknown way [117,118]. Infusion of 20β-OH-ecdysone to mice for five days fostered the increase of triceps mass (115 mg) compared to the control groups treated with placebo saline solution (88 mg; *p* < 0.01). Further microarray RNA analysis exhibited alterations in genes associated with skeletal muscle cells development, growth, and morphogenesis (*Gdfs*, *Gdf5*, *Tor2a*, *Pten*, *Traf6*), but after false discovery rate (FDR) correction, no significant difference has been detected in the expression level of these genes between the treated and the control animals [86]. An *in vivo* study of 20β-OH-ecdysone transformation in mammalian system revealed that the main metabolic product, poststerone, also fosters the increase of muscle type fiber cross-sectional area but in a relatively lower degree compared to its parental compound *per se* [119]. However, this evidence emphasizes the putative anabolic property of ecdysterone caused by the steroidal nucleus of this compound.

Eventually, based on the body tissue analysis (muscle cells growth, an increase of fibers cross-sectional area) and known aspects of the protein synthesis gain, most authors proposed that benefits of 20β-OH-ecdysone are caused by the activation of the PI3K/Akt pathway. It seems engaging, but the protective effect of 20β-OH-ecdysone has also been reported in muscle atrophy rats. It was suggested that the result is associated with inhibition of ubiquitination (Table 1), but the exact molecular mechanism of action was not proposed [87] (Table 1).

Due to possible androgenic and growth-stimulating effects, 20β-OH-ecdysone has been examined on the subject of acceleration of bone mass augmentation. Mice treated with 20β-OH-ecdysone at a dose of 0.5 mg/kg for three weeks with a frequency of 5 times per week showed an increase of strength and volume of both trabecular and cortical bones up to 0.88 ± 0.03 and 0.79 ± 0.2 (*p* < 0.05) in male and female animals for trabecular bone volume, and up to 15 ± 0.3 and 12 ± 0.5 (*p* < 0.05) in male and female animals for cortical bone volume [120]. Although the bone mass augmentation is pre-determined by gender-specific hormonal background and, specifically, by estrogen receptors loading, authors demonstrated that 20β-OH-ecdysone provides equal impact regardless of the gender feature (Table 1). Earlier, experiments with ovariectomized rats fed with ecdysone over three months also showed a positive effect on bone mineral density, which was reduced almost at half after controlled treatment [121,122].

The protective effect of 20β-OH-ecdysone against the loss of bone volume and density due to competing interaction with glucocorticoids and inhibition of stromal cells autophagy has been observed in rats who were being implanted with slow-releasing (for 60 days) prednisolone [120]. Regular treatment with 20β-OH-ecdysone entailed to the elevated level of bone mineralizing surface (up to 43%, *p* < 0.05) and the increased bone formation at the endocortical surface (up to 213%, *p* < 0.05). Oddly, but such long-term regular treatment with 20β-OH-ecdysone did not induce alteration in the basal levels of the hormonal background, including insulin, leptin, estrogen, T3, and thyroxine, which suggests the targeted impact of this sterol (and products of its transformation) on the differentiating osteoblasts.

Nevertheless, despite many indications about the growing rate of protein synthesis under the regular administration of 20β-OH-ecdysone, the assumption regarding its direct influence on the muscle mass gain and activation of satellite cells seems to be inconsistent and out of proportion because many suggested mechanisms do exist under the massive revision (activation of sex hormone receptors [116,117], stimulation of Akt signaling [88], stressing of satellite cells [118] and release of calcium through the action on G protein-coupled receptors [123], inhibition of ubiquitin-proteasome machinery, etc. However, unfortunately, neither yielded strong evidence to endorse any of these ideas. The tacit assumption is that the anabolic properties of 20β-OH-ecdysone are mediated by its structural similarity with mammalian sex hormones. Admittedly, it is still a suggestion since no one tested whether 20β-OH-ecdysone does alter the sensitivity of estrogen/testosterone receptors. The possible implication of estradiol receptors rather than androgen receptors activation has been previously proposed based on the in silico molecular docking [116]. However, the molecular model comes at odds with the results of a fit-for-purpose experiment, which displayed that regulatory activity of 20β-OH-ecdysone bypasses this way of interaction, and it does not bind with estradiol receptors [122].

The research purposed to determine the influence of plant sterols on the excretion rate and to determine that the origination of excreted anabolic steroids (in particular, boldione) demonstrated the presence of anabolic steroids in both control and tested groups who were administrated with plant sterols. However, no significant correlation was established between the excretion rate of 17-ketosteroids precursors (for example, androstenediol) and the consumed amount or the frequency of plant sterols intake [68]. Earlier, these authors and other research groups reported about *in vitro* transformation of the selected plant sterols into androgenic products such as testosterone and androstenedione. In those cases, the observed phenomenon has been proposed by the activity of residential gastrointestinal microflora [124,125,126].

The molecular mechanism responsible for the transformation of sterols has been suggested after discovering at least three genes (*kstD1*, *kstD2* and *kstD3*) encoding specific FAD-dependent dehydrogenases (EC 1.3.99.4) in *Rhodococcus erythropolis* strain SQ1 [127,128,129]. The proposed transformation is composed of several consequent reactions, including oxidation of 3-hydroxyl-5-ene core of sterols to 4-ene-3-one core and, eventually, generation of testosterone and androstenedione [127,128]. These enzymes are capable of introducing a double bond between C1 and C2 atoms of steroidal nuclei and display different substrate preferences and catalytic property, but all can catalyze the transformation of both saturated and unsaturated steroids to 3-keto-1-ene steroids [130,131]. Biotransformation of plant sterols to 17-ketosteroids (mostly to 4-androstene-3,17-dione) by soil bacteria, intestinal microflora, or plant-associated bacteria has been repeatedly demonstrated as a reaction of sterols’ C24 side-chain cleavage and consequent conversion of 3-oxo-24-ethylcholest-4-en-26-oic acid into 3-oxochol-4-en-26-oic acid [132,133]. Therefore, it has been proposed that such kind of transformation can regularly undergo in the mammalian intestine because of the ubiquitous prevalence of plant sterols in food products [134], including snack bars [135,136], soft drinks [137], and oils [138]. Accordingly, there are enough substrates accommodated for in the intestinal microflora and that are capable of being accumulated in a time- and dose-dependent manner for further transformation, and the reaction can be accelerated and improved if the diet is supplemented by prebiotics.

By no means, it does not make plant sterols a foe of humans because there are many more benefits of phytosterols that provide a positive sensation in *post hoc*. As has been touched on above, plant sterols can be secreted from the intestine into lymph and bloodstreams within chylomicrons and HDL, whereupon liver cells trap them and incorporate into the bile acids synthesis pathway [139]. Yet, as far-fetching as that seems, some plant sterols (sitosterol, campesterol, stigmasterol, episterol) can be utilized as precursors for the endogenous synthesis of human 17-ketosteroids, including DHEA (dehydroepiandrosterone), which is catalyzed by the congregative action of lathosterol oxidase, 3β-hydroxysteroid dehydrogenase, Δ24-sterol reductase, Δ7-sterol 5-desaturase and cytochrome CYP51A1 [140,141,142]. Some species of *Mycobacterium* are capable of using plant sterols as a substate for producing and accumulating androstenedione and androstadienone, albeit these steroids can be toxic for the host-microbial cell due to weak solubility in water [89].

The impact of plant sterols on the level of serum testosterone has been revealed in animals who were intentionally treated with functional food enriched with β-sitosterol. Besides, mice of the F2 generation were featured by a higher testicular level of testosterone, whereas female mice had a higher plasma level of estradiol [77]. Nieminen et al. also reported that the level of sex steroids, including estradiol, substantially increased after controlled treatment with plant sterols; however, no significant effect was observed for the rate of secretion of pituitary hormones and the serum level of cholesterol [76,143]. Considering evidence-based data, authors assumed that observed alterations in the level of sex hormones were not receptor-mediated but rather were stipulated directly by plant sterols *per se* that might take the role of steroid precursors. Although the suggestion sounds plausible, the authors did not observe changes in the activity of enzymes implicated in the synthesis of steroids or the transformation of cholesterol.

The occurrence of anabolic steroids in plants was rigorously reviewed by [144], who documented that the prevalence of progesterone (up to 80 ng/g) was in more than 80%, and the prevalence of androgens (androsterone up to 11.4 ng/g, testosterone up to 80 ng/g, and epitestosterone up to 110 ng/g) was in more than 70%, and 17β-estradiol (in amounts up to 40 pg/g) was in nearly in 50% cases among 128 tested plant species of 50 families. Most of the detected and reported steroids were reported as not being contaminants of plant subjects, but instead, they were substrate-specific intermediates that naturally occur in plants. For example, progesterone found in *Digitalis lanata* is a necessary compound for the production of cardiac glycosides biosynthesis, which is accomplished by ∆5-3β-hydroxysteroid dehydrogenase (EC 1.1.1.145), which converts pregnenolone to isoprogesterone (Figure 3) [145]. Treatment of some plants with steroid biosynthesis inhibitors results in the abrogation of flowering on 85%, which indicates the endogenous origination and obligatory role of some steroids for plant development [146]. While the selected estrogens stimulated flowering [147], androsterone and androstane, oppositely, did not produce the expected effect. The positive effect of estrogen can be caused by an estrogen-binding protein found in plants and shared structural similarity with the mammalian estrogen receptor α-subtype. This protein was detected in the nuclear fraction of *Solanum glaucophyllum* and *Lycopersicon esculentum* as a 67 kDa compound using highly specific antibody analysis. Unexpectedly, but protein localized in this fraction was able to bind 17β-estradiol with high specificity and selectivity according to ligand blot analysis [148,149].

It should be admitted that the inspected effect might be caused by phytoestrogens as secondary metabolites of plant sterols, which structurally and functionally resemble human estrogens. However, the presence of phytoecdysteroids, as broadly discussed above, in many plant species, despite its natural origination in insects, enforces to keep the discussion about the possible biosynthesis of human sex steroids and their crucial role in plants seriously.

Some plant extracts, for example, *Tribulus terrestris*, widely applied as food additives in sport, are capable of managing steroidal balance in mammalians. Bolus administration of this extract at doses between 7.5 and 30 mg/kg to *Macaca mullata* resulted in a meaningful increase (up to 52%) of serum levels of testosterone, dihydrotestosterone, and dehydroepiandrosterone sulfate (DHEAS) [150]. The obtained results are quite confident because the measurement of endogenous steroids was elaborated with the use of tritium-labeled internal standards ([^3^H]-testosterone, [^3^H]-dihydrotestosterone and [^3^H]-dehydroepiandrosterone sulfate). Authors showed that even a minimal dose (7.5 mg/kg) of *Tribulus terrestris* induced the rise serum level of dihydrotestosterone already in 15 min after intravenous administration, whereas serum level of DHEA delayed and increased more gradually 60–180 min after administration. On the contrary, later research with human subjects, who were orally administrated with capsules of *Tribulus terrestris* at a dose of 1250 mg/capsule and received high-volume and high-intensity training for three weeks (separate by four weeks of rest), did not demonstrate a significant effect on muscle mass gain. There were marginal changes in levels of serum testosterone (15.4 ± 6.8 mmol/L in baseline and up to 16.9 ± 7.1 mmol/L, *p* < 0.05, in trained subjects both, administrated with the extract) and dihydrotestosterone (880.3 ± 348.9 ng/mL in baseline and up to 774.1 ± 272.0 ng/mL in trained subjects, *p* < 0.05) [151].

Meanwhile, it was found that the serum level of insulin-like growth factor-binding protein 3 (IGFBP-3) substantially decreased in subjects, who ingested *Tribulus terrestris* during training, but no correlation was found between the IGFBP-3 alterations and muscle mass. So, the net benefits of the extract were almost neglectable, and anaerobic performance was the only possible improvement that can be customized after long-term administration. Though it seems somewhat unexpected since the increase of muscle regeneration capacity and protection of unloaded muscles are known and well-characterized modulating effects of IGFBP-3 and IGF-1, which are generally accompanied by alterations in the level of sex hormones and their transformation products.

Detection of androstadienone and androstenedione at a concentration of 2.20 ng/g has been reported for *Nicotiana tabacum* [152], pine pollen [153], and even in potato, where marginal concentration did not exceed 50 pg/g [154]. Yet in the 1970s and 1980s, there were experiments showing that [^14^C]-labeled androstenedione can be converted to testosterone in cultivating seeds of *Nicotiana tabacum* [155]. There are many contradictory reports, but it seems on the surface that plant sterols may produce a positive effect on human performance. The exact mechanism is mostly unknown, but it can be customized and be sterol-type-specific. It is hardly believed that plant sterols are directly introduced in mammalian steroidogenesis and can be transformed in either of sex hormones. There is evidence about the shared structural similarity of estrogen-binding receptors between mammalians and plants but no strong chemical and biochemical proofs about the existence of estrogens endogenously synthesized in plant organs. Thus, the assumption about the positive effect of phytoestrogens rather than plant-derived 17β-estradiol sounds more plausible and reasonable. Eventually, several plant enzymes that are putatively involved in the synthesis of mammalian-like sex hormones were found, but the majority of results are on the level of mechanical assumptions. However, one research even proposed the possible mechanism of unusual steroidogenesis in plants that involves progesterone 5α-reductase, ketoisomerases, and dehydrogenases to produce progesterone and corticosteroids [156], and appropriate receptors relevant to these steroids were detected in cytosolic and microsomal fractions of plant organs [157,158]. The crucial role and necessity of such mammalian-like hormones in plants are debatable, but it is believed that, for example, progesterone can be involved in stress response and regulation of growth in plants while estradiol may affect plants flowering.

## 6. Fake Sterols and Contaminations

Due care must be taken when consuming plant-derived nutritional supplements that promise legitimate improvement and enhancement of human abilities. In 2008 and recently in 2020, the anti-doping laboratory in Germany conducted a specially designed study among a vast number (634 in total) of nutritional supplements purchased in 13 different countries and showed that they contained up to 15% of non-declared prohibited substances, including synthetic anabolic steroids and so-called “design” steroids [159,160]. Thus, the anabolic effect of some sterol-based food additives and supplements can be faked by intentional cross-contamination and adulteration with known drugs of abuse, including non-natural prohormones.

However, risk factors are associated not solely with food supplements. Evidence exists about entering the naturally occurring sex hormones in plants through soli, reclaimed water, and water environment after human excretion [161] since synthetic and natural steroids are generally used in pharmacy as a compound of contraceptives, anti-cancer drugs, and cosmetics [162]. It has been estimated that the total rate of steroids excretion, for example, in Chinese rivers can reach up to 370 tons per year, of which about 1.3 tons are wasted in the ocean system annually [163]. Yet, in 2012 National Cancer Institute provided analysis of stream waters in 14 sampled in 14 states of the USA and observed pollution by glucocorticoids and androgen steroids in 27% and 35% of the analyzed samples, respectively [164]. Treatment of plants by sewage waters in Tunisia displayed strong androgenic and estrogenic activities caused by the presence of endocrine-disrupting compounds (glucocorticoids, pregnenolone, progesterone, aldosterone) in water [165]. The assay supported by the gas chromatography–mass spectrometry (GC-MS) of plants treated by municipal wastewaters in Australia revealed up to 356 ng/L of estrogen equivalents in plants leaves [166]. These steroids can be taken up by and accumulated in plants and further recycled by soil bacteria. Their total accumulated content may vary between 45 ppm to 168 ppm depending on sterol’s side-chain moiety and solubility property affected by soil type and water pH (Figure 4) [167,168].

Experiments with tightly controlled growth of lettuce plants (*Lactuca sativa*) showed that the plants contain enough of 17β-estradiol as a natural human steroid and ethinylestradiol as a drugs-derived synthetic steroid. It was demonstrated that both compounds are accumulated primarily in the root (13%) and less in leaves of lettuce (10%). Besides, it was established that 17β-estradiol inclined more for accumulation and uptake compared to ethinylestradiol, and its content after the first week of controlled growth was only 3.07% and 0.54% in roots and leaves, correspondingly [169]. However, at the end of the last week of growth, the content of ethinylestradiol exceeded 12% and 8% in roots and leaves, correspondingly. The high bioavailability of estrogens by plants is supposed to be due to their moderate hydrophobicity; thus, they are more tending to be soluble in water and being taken by plants [168,170]. For instance, ethynylestradiol can adversely affect brain operation and behavior pattern. The increasing damage of the male and female reproductive function can also be associated with hormone pollution taken with food and water. The risk associated with a daily intake of steroid-enriched food only can be considered if the total content exceeds 3 μg/day according to recommendations of the WHO committee on food additives, but typical concentrations are ranged between 110 and 400 ng/g of plant or algae weight depending on the environmental condition, area of growing, manufacturer location, water pollution level, and density of human population in the nearest area [171,172,173].

The main message coming from these investigations is that the abundantly detected in plants human steroids (i.e., estrogens and androgens) can be accounted as accidental contaminations entering plants from soil and water sources rather than their endogenous origin. Their stimulating impact on plant growth, germination, and flowering can also be caused by binding with phytoestrogen receptors due to structural similarity but with lower affinity than the native endogenous plant sterols. Exogenous steroids that enter a plant’s metabolism may also influence the balance of inorganic compounds and the depending biochemical reactions such as redox reaction, photosynthesis, osmotic regulation, and setting of ions balance [174]. Obviously, the exogenous steroids would influence plants the same way as plant sterols are metabolized by mammalians but in a very limited fashion. Entering mammalian steroids into plant metabolism is enabled due to moderated hydrophobicity for some of them, but their metabolic pathways are poorly scrutinized and mostly limited to assumptions and are therefore scarcely even meaningful. Unless there is no evidence about the toxic effect of plant sterols on humans, plants are known to be vulnerable to the impact of mammalian sex hormones. Some sex steroids may negatively impact plants due to their accumulation in plant parts, metabolic switching, and low solubility in the cytosol.

## 7. Conclusions

Starting from the year 2010, a reader can find in PubMed library more than 4000 reports about plant sterols metabolism and their benefits for human health, and the number of such reports increments at least 15% annually. One can say that it indicates continual and growing interest in sterols as an inexhaustible source with great potential for medicine, food industry, pharmacy, and cosmetics. Having begun in 1954 from the report about the cholesterol-lowering effect and up to nowadays, sterols have become widely distributed as our friends in food additives and pharmacy, and as alerting objects in professional sport. There is a massive number of publications about the natural occurrence of mammalian-like steroids in plants and the possible transformation of plant sterols to sex steroids in mammalians. Still, such debatable information must be perceived with due criticism and caution. While some researchers report about estrogen and androgen receptors in plants, positive effect on germination and endogenous synthesis of progesterone, testosterone, and other steroids in plants as obligatory intermediates that are necessary to produce end-point phytoestrogens and other sterols, other research groups continue to oppose them and argue about entering steroids through reclaimed water and soil pollution, lack of direct effect for plants growth or the biochemical inability for steroids producing in plants or, oppositely, the transformation of sterols in human steroids.

It seems that the overall effect of plant sterols on human health is rather positive since there is no rich information about their toxicity apart from the ambiguous anabolic effect for some of them (Figure 5). However, even the net advantages of functional food and additives are somewhat questionable because the effects of muscle mass gain and glycolysis metabolism are not always meaningful or unexpectedly marginal.

Still, plant sterols application and utilization in a variety of field are of growing and paramount interest for the food industry since we intake sterols every single day and favorable properties of sterols for human health (such as anti-aging, anti-cancer, antioxidant, anti-atherogenic, and many other anti-”bad” items) are the main alarming signatures calling the industry and food market to elaborate under new dietary products. Various recent research and clinical trials, including those mentioned in this review, evident that a plant-based diet is the most appropriate and accessible way to protect our health, raising the popularity of “green food” over the world. Numerous publications about cardioprotective and anti-aging properties of some phytochemicals played, most likely, the central role in considering the association between food and benefits for human health.

However, many controversial properties of plant-based food, including being enriched by plant sterols, can be captured by interindividual variability greatly described when talking about plant sterols effect. Obviously, as many people you would ask about their dietary preferences, as many diet patterns you will learn about. While keeping in mind the uncertainty of plants sterols metabolic transformation in human, the individual, or personal, response to bioactive food will be the most important determinant in elaboration of specific diets or food intake schemes. So far, the exact effect of plant sterols is poorly predictable and requires a deep investigation of nutrigenomic, epigenetic and even ecological factors prior to plan and develop diet recommendations purposed to improve abilities in sport or avoid cardiovascular complications, or retard skin aging, etc. Consequently, a very careful personalized approach based on the highly-specific bioactive compound, including a specific sterol, should be developed, while considering different nutritional needs and different categories of population.

However, until we do not know for sure what the reasonable long-term effects of plant sterols on our life are, or whether plants can produce steroids or sterols can be transformed into steroids, it would be appropriate to keep in mind and follow the rule coined by Ludwig Feuerbach (1804–1872): “Man is What He Eats.”

## Figures and Tables

**Figure 1 nutrients-13-01623-f001:**
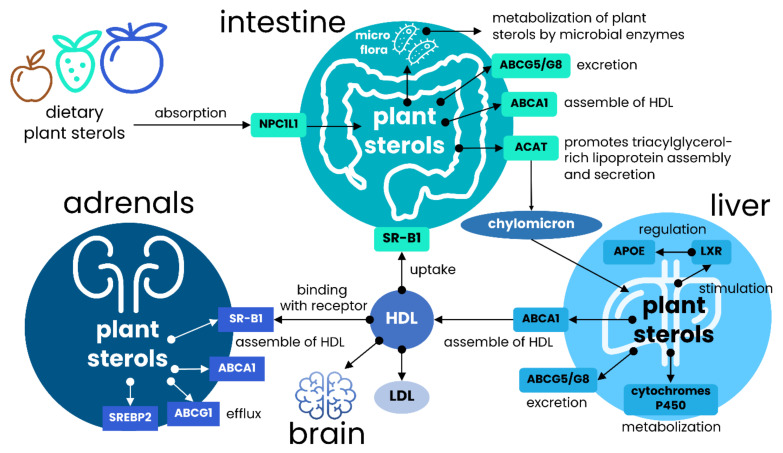
General scheme of the major routes of plant sterols in human. Plant sterols and dietary cholesterol are mainly absorbed in intestine, and Niemann-Pick C1-like 1 (NPC1L1) transporter plays a vital role in the regulation of initial absorption. After absorption, sterols are esterified in support with ACAT and transported to the liver following the incorporation into chylomicrons. In contrast, unesterified sterols are pumped out by ABCG5/ABCG8 transporters, which are the major cholesterol and plant sterols transporters. The plasma efflux of both plant sterols and cholesterol is regulated by ABCA1, which is involved in the assembling of HDL-like particles with the assimilated sterols. This transporter is also critically important in plant sterols and cholesterol efflux after delivering chylomicrons from enterocytes to liver cells. Plant sterols are capable of stimulating LXR receptors regulating APOE expression, which is essential for HDL and LDL assembly and uptake; and can be partially catalyzed by microsomal cytochromes. Binding with LXR receptors upregulates ABCG5/G8 transporters, and thus enhances cholesterol and plants sterols absorption. The exported HDL-like particles with the incorporated plant sterols are trapped by SR-IB receptors expressed on the liver and adrenal glands cell surface. This receptor plays a pivotal role in the uptake of cholesterol through HDL and, most importantly, for brain lipids metabolism, where HDL is the primary source of lipids and cholesterol uptake.

**Figure 2 nutrients-13-01623-f002:**
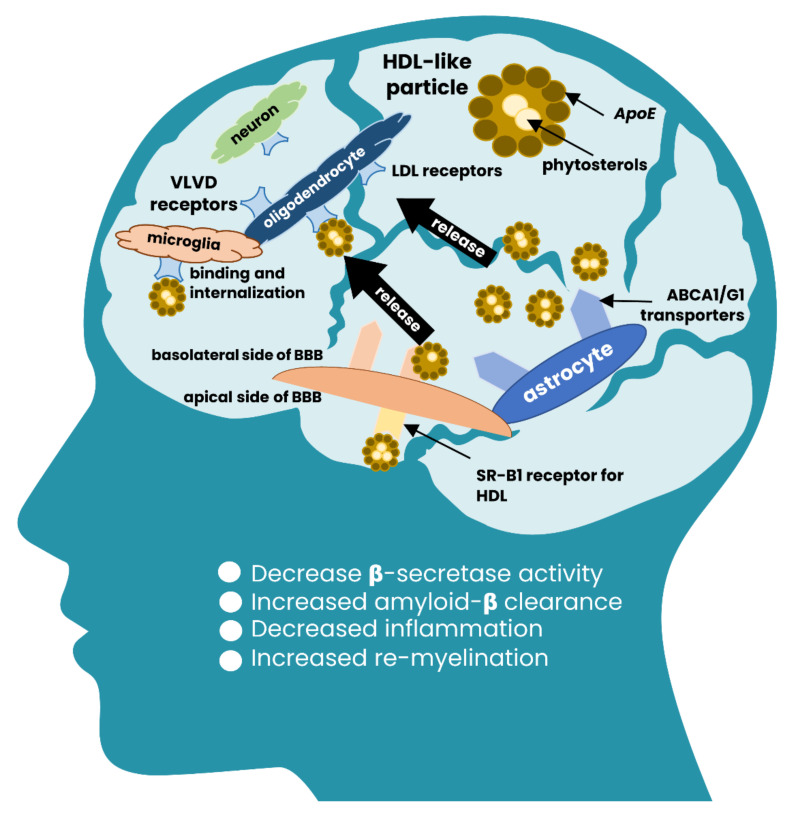
The proposed scheme of plant sterols entering the brain system. The mechanism involves HDL-like particles carrying plant sterols and apolipoprotein E (APOE). The HDL-like particles are translocated through SR-B1 HDL receptors on the apical side of the blood-brain barrier (BBB) and released into the brain via ABCA/ABCG1 transporters, which are expressed on the basolateral side of BBB and on the surface of astrocytes. LDL/VLDL receptors trap the released HDL-like particles enriched with plant sterols on the surface of microglia and oligodendrocytes since HDLs associated with APOE are the primary source of lipids transport in the central nervous system. Upon delivery, plant sterols activate mechanisms that decrease inflammation, increase amyloid-β clearance, and reduce the activity of β-secretase. The regulatory mechanism is largely unknown but is supposed to be realized through the activation of LXR/RXR receptors and regulation of APOE expression mediated by PPAR-signaling.

**Figure 3 nutrients-13-01623-f003:**
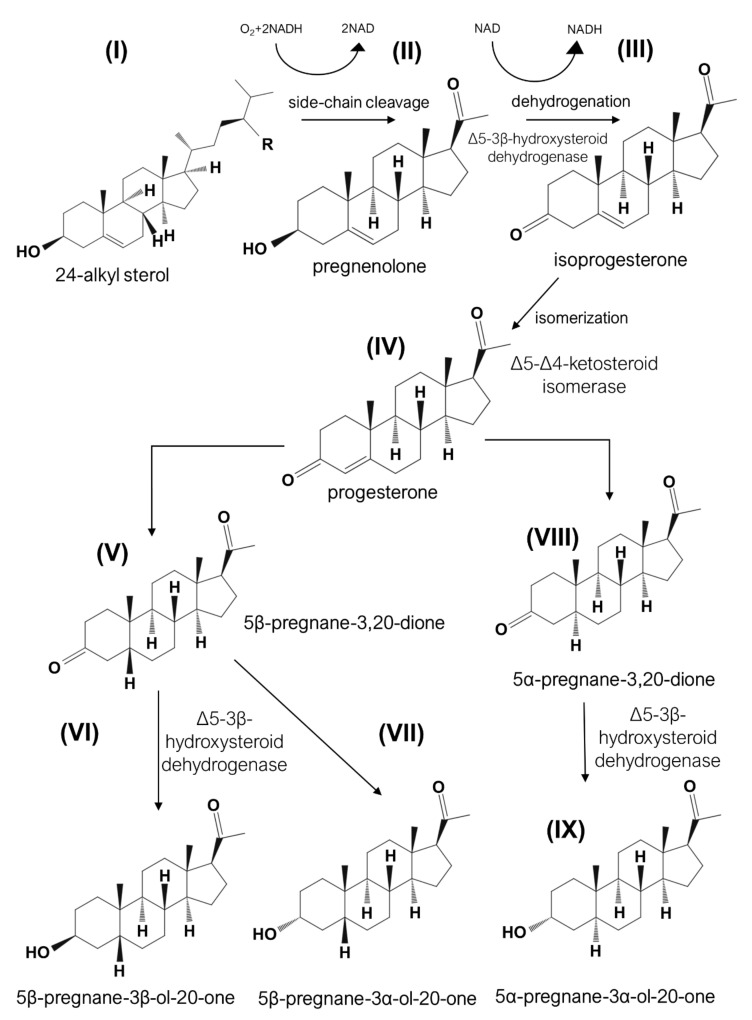
The proposed scheme of biosynthesis of progesterone and pregnenolone in plants. It is assumed that the synthesis starts from the cleavage of 24-alkyl sterols (I) side-chain resulting in the production of pregnenolone (II) following dehydrogenation in the presence of Δ5-3β-hydroxysteroid dehydrogenase (EC 1.1.1.145) activity. The resulting isoprogesterone (III) is further isomerized by Δ5-Δ4-ketosteroid isomerase (EC 5.3.3.1) to progesterone (IV). Further transformation to products (V)-(IX) is accomplished in the presence of Δ5-3β-hydroxysteroid dehydrogenase (EC 1.1.1.145) activity that, eventually, catalyzes the production of glycosides. It is assumed that progesterone and pregnenolone are essential intermediated in plant biosynthesis of glycosides. Both enzymes are ubiquitously distributed among plants and microorganisms, whereas in animals, these two enzymes reside in one-single protein. The Δ5-3β-hydroxysteroid dehydrogenase has a higher preference for NAD instead of NADPH, which is also accepted by lower activity.

**Figure 4 nutrients-13-01623-f004:**
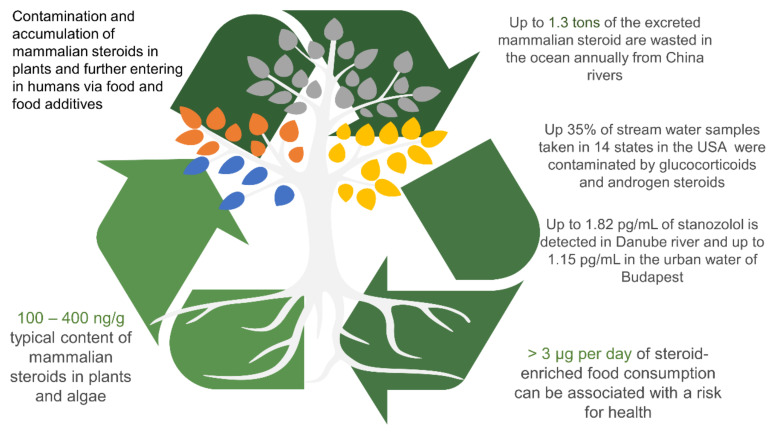
It is hypothesized that mammalian-like steroids found in plants are contaminants that enter the plant through the soil and reclaimed water. Up to 1.3 tons of the excrete mammalian steroids (endogenous and synthetic) are wasted in the ocean from China rivers. In the USA, glucocorticoids (hydrocortisone and prednisone) and androgen steroids were found in 27 to 35% of stream water samples taken from 14 states. These polluting steroids enter plants, fungi, and soil microbial cells and can be highly toxic for plants due to low solubility. The absorbed steroids can be accumulated in plant organs (roots, leaves) and consumed by humans. According to the WHO recommendations, the risk associated with steroid-enriched food consumption is considered only if their intake exceeds 3 µg per day. However, it is believed that the specific content of contaminating steroids in plants is 100 to 400 ng/g of plant weight.

**Figure 5 nutrients-13-01623-f005:**
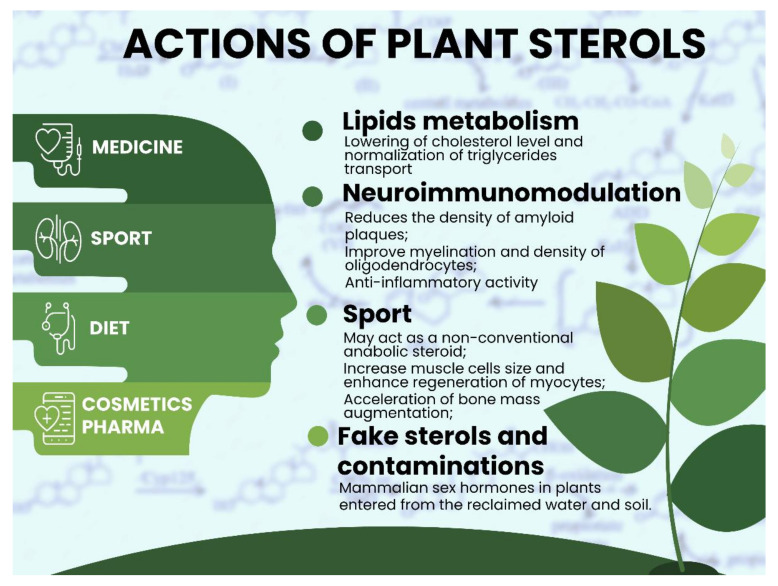
The overall effect of plant sterols and, in particularly, functional food is rather positive. Up to nowadays, sterols are widely used in different ways, but their natural effect (such as implication in lipids metabolism and neuro- and immunomodulation) does merit attention and seems to be vital for reason of their steroid-like properties. Therefore, it is not surprising that sterols are used in cosmetics, pharma and sometimes are considered in professional sport as suspicious substances that may improve human abilities.

**Table 1 nutrients-13-01623-t001:** Main findings of plant sterols effects in clinics and professional sport.

Type of Study	Characterization and Study Design	Purpose of Study	Observed Effect	Reference
Review study	Food enriched by plant sterols and stanols	Evaluation the contradictory effects of plant sterols on cholesterol lowering	Decrease of LDL-C by 10%	[33]
Meta-analysis	Data of clinical trials between January 1992 and April 2013	Evaluation of plant sterols delivery way and matrix on cholesterol-lowering action	Decreasing of LDL-C by 12 mg/mL for 4–6 weeks; no difference between sterols supplements and food enriched by plant sterols	[38]
Randomized controlled clinical trial	Supplementation by sterols-enriched margarin	Effect of chronic intake of plant sterols on postprandial metabolism	Lowering of total cholesterol by 7.3%, LDL-C by 9.5% and LDL by 12.3% after intake of sterols-enriched margarine for 3 weeks	[53]
Randomized controlled trial	In vivo trial with both men (*n* = 6) and women (*n* = 6) for 5 weeks after consumption of phytosterols	Endogenous origin of boldenone and boldione under controlled phytosterol-enriched diet	Endogenous boldione produced between 0.75 and 1.73 ng/mL and strongly correlated with the consumption of plant sterols; in contrast, the endogenous origin of boldenone was not confirmed under the same condition	[68]
Randomized controlled clinical trial	The study on 28 subjects with hypertriglyceridemia (elevated fasting triacylglycerols) supplemented by stanols-enriched (2.5 g/day) margarine for 3 weeks	Effect of plant stanols supplementation on lipids metabolism	Plant stanol esters supplementation (margarines) lower triacylglycerol by 6.7% and LDL-C by 9.5% after 3 weeks intake	[69]
Clinical trial	Women with coronary artery disease consumed margarine without and with sitostanol (3 g/day) ester for 7 weeks	Effect of sitostanol-enriched diet on serum level cholesterol in postmenopausal women with myocardial infarction	Lowering of serum cholesterol by 13% and LDL-C by 20% for 12 weeks in home diet and by 11% and 16%, correspondingly, if combined with simvastatin	[54,67]
Comparative study; human subjects	The study on 144 subjects aged 30–80 years old	Age-related relation between some sterols and cognitive performance	The level of cholesterol, desmosterol, sitosterol and campesterol were not related to cognitive performance; however, serum levels of lathosterol and lanosterol inversely correlated with cognitive performance	[66]
Comparative study; animal model	Hypercholesterolemic (ApoE^−/−^) mouse with acute aseptic was fed with 2% plants sterols supplements	Investigation of immune modulating and stimulatory effects of plant sterols	Increased IL-6, IL-2 and IFN-γ secretion in spleen cells; potentiating T-helper response	[61]
Comparative study; animal model	Fibroblast and keratinocytes cell culture and mouse model of cutaneous wound healing	Effect of homobrassinolide on proliferative activity and migration of mouse fibroblasts and keratinocytes	A 2-fold increase rate of wound closure if mice received 10 µg topical brassinosteroids; suppressed ICAM-1 and TNF-α but weak effect on TGF-β; upregulation of Akt-1 phosphorylation in treated mice	[64,70]
Comparative study; animal model	Transgenic mice with disrupted BBB caused by lacking the PDGF-B retention motif	Metabolic circulation and sterols flux under the condition of disrupted blood-brain barrier in pericyte-deficient mice	A significant accumulation of campesterol and sitosterol in the brains but the degree of accumulation of sitosterol was lower; higher mRNA levels of HMG-CoA synthase. Significantly increased flux of cholesterol from circulation into brain	[71,72]
Comparative study; animal model	Mice deficient for ATP-binding cassette transporter G5 (*abcg5*)/G8 (*abcg8*) and deficient on *ApoE*	Association of plant sterols and cholesterol brain concentration with ApoE and transporters activities	Increased serum level of plant sterols (7–16-folds change); upregulation of *ApoE* mRNA level	[73]
Comparative study; animal model	LXRβ^−/−^ and wild type study mice treated by β-sitosterol for 3 weeks at a dose of 42 mg/kg	Toxicity effect of β-sitosterol on neurons	Sever symptoms of paralysis and dopaminergic disfunction in transgenic mice; aggregation of ubiquitin in the cytoplasm of large motor neurons; increased level of 24-hydrocholesterol. Wild type mice were not affected	[74]
Comparative study; animal model	Loading of 7β-hydroxysitosterol in ApoE^−/−^ mice for 28 days	Sterols and cholesterol absorption rate in plasma, liver and brain of transgenic ApoE-deficient mice	Concentration of 7β-hydroxysitosterol increased 65.8-folds and 21-folds in brain and plasma, correspondingly	[75]
Comparative study; animal model	European polecat (*Mustela putorius*) fed by phytosterols	Endocrine and metabolic effects of plant sterols	Increased plasma estradiol, TH and glucose-6-phosphatase activity, the plasms ghrelin level decreased; the total serum level of cholesterol was not affected	[76,77]
Prospective study; anima model	Examination of brain cortex and hippocampus in 6-, 12-, 18- and 24-month-old rats	Age-related changes of plasma and brain plants sterols concentrations	Reduction in lanosterol (by 28%), lathosterol (by 25%) and desmosterol (by 51%) concentrations at 24 months in the cortex and hippocampus that can be caused by the loss of synaptic plasticity	[65]
Research study; animal model	Murine colitis model fed by sterols-enriched food at a dose of 400 mg/kg per day	Effect of plant sterols mixture on gastrointestinal inflammation	Low level of leucocyte infiltration into the colon, preserved epithelial integrity; accelerated reparation of epithelial structure	[78]
Research study; cell culture	In vitro fibroblast L929 wound healing assays exposed to plant leaves ethanolic extract at a dose of 3 µg/mL	Investigation of wound healing effect of ethanolic extract of *Wedelia trilobata* (L.) leaves	Stimulatory effect; increased collagen content, activity against gram-positive bacteria, increased survivability by 85% after oxidative stress; induced by 70% migration rate	[79]
Research study; animal model	Rat model of Parkinson’s disease induced by β-sitosterol derivate for 16 weeks at a dose of 3 mg 5 days per a week	Effect of Beta-sitosterol beta-d-glucoside on dopaminergic neurons	ß-sitosterol may produce toxic glycosides and modulate lipids metabolism in neurons; induces the loss of dopaminergic neurons of the nigrostriatal pathway and results in akinesia and loss of locomotion	[80]
Research study; animal model	Male APPswePS1ΔE9 Alzheimer’s disease mice supplemented with 50% (*w/w*) pulverized dried *Sargassum fusiforme* diet	Effect of sterols on cognition ability through the activation of LXRβ receptors	Extracts of *Sargassum fusiforme* activates LXRβ at a dose 5 ug/mL and markedly decreased Aβ plaque load in the cortex (70%) and hippocampus (81%)	[81]
Research study; animal model	Animals with experimental autoimmune encephalomyelitis	Evaluation of motor skills performance and relationship between cholesterol and myelination in cuprizone model mouse fed by phytosterols diet	Declined expression of IFNγ, TNF and IL-17 and associated decrease of serum cholesterol for mice treated with 5% w/w dietary cholesterol	[82]
Research study; animal model	Type 2 diabetes mellitus rats	Effect of *Aloe vera* gel on expression level of PPAR targets	Activation of PPAR and decrease of Apoc3 in dose-dependent manner	[83]
Research study; animal model	Rats with non-alcoholic steatohepatitis daily administrated with *Aloe vera* extract (50 mg/kg) for 8 weeks	Effect of *Aloe vera* on liver enzymes and inflammation in non-alcoholic steatohepatitis model rats.	PPAR decreased; IL-18 and caspase-3 increased in rats fed with *Aloe vera*. Symptoms lobular inflammation and hepatocyte apoptosis were attenuated if animals treated by *Aloe vera*.	[84]
Research study; cell culture and animal model	Murine embryonic fibroblast cell line CCL-92 exposed to brassinosteroids and its synthetic analogues; topical application of synthetic brassinosteroids on mice after injury	Stimulating effect of homobrassinolide and its analogues on cell fibroblasts and keratinocytes proliferation and migration; and in vivo wound healing in animals	Brassinosteroids promotes cell migration into a wound zone with an efficacy of 30% at 5 μM after 12 h of incubation; treatment with 1–10 μM resulted in an increase in the proliferation of primary keratinocytes	[64]
Research study; cell culture	Rat L6 skeletal muscle cell line CRL-1458 treated by brassinosteroids analogues	Evaluation of anabolic effect on muscle cells through the selective activation of Akt by different brassinosteroids	Decrease in protein degradation by 20% after treatment with 10 uM of brassinosteroids in a dose-dependent manner	[85]
Research study; animal model	Mice (12 weeks old) strain C57BL/J6 were continually administrated by 20-hydroxyecdysone at a dose of 5 mg/kg/day	Anabolic effect of 20-hydroxyecdysone administration	Increase in the mass of the triceps brachii by 30%; however, no differentially expressed gene were observed compared to the control group (saline) after applying FDR. In summary, administration of 20-hydroxyecdysone did not demonstrate the pronounced anabolic effect	[86]
Research study; animal model	Adult male Wistar rats (10 weeks old) with tenotomy-induced muscle atrophy administrated 20-hydroxyecdystone at a dose of 5 mg/kg for 7 days	Evaluation of 20-hydroxyecdystone type-specific response in atrophy or injured muscles	Administration of 5 mg/kg of 20-hydroxyecdystone tended to alleviate tenotomy-induced reduction of muscle mass and attenuate by 63.1% tenotomy-induced ubiquitinated protein in soleus muscle, but had no effect on cross-sectional area in atrophied muscle	[87]
Research study; cell culture	Mouse skeletal muscle cell line C2C12 treated by 1 µM 20-hydroxyecdyson delivered in ethanol	Effect of 20-hydroxyecdyson on calcium influx and Akt activation	Treatment with 1 μM 20-hydroxyecdyson increased intracellular Ca^2+^ already in 35 sec and activated Akt more than 3-fold in 2 h; however, pretreatment with phospholipase C inhibitor abolished the effect of 20-hydroxyecdyson	[88]
Biotechnology and bioengineering	Mutants of *Mycobacterium* spp.	Investigation of transformation of plant sterols into androstenedione and androstadienedione in bacteria	Bacteria, growing on phytosterols, accumulate androstenedione and androstadienedione as the steroid intermediates during biotransformation of plant sterols after removing the side-chain and ring cleavage	[89]

## Data Availability

This is a review paper, although all presented data are available in the appropriated original cited papers.

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
