# Peer review of "Diversity of Plant Sterols Metabolism: The Impact on Human Health, Sport, and Accumulation of Contaminating Sterols"

_nutrients, 2021, doi:10.3390/nu13051623_

Round 1

Reviewer 1 Report

My major concern is the low quality of the Figures, making it difficult to see, and the letters are very small in Figure 1.  I think that in general it is an interesting topic and that it is well written. Perhaps, the conclusions could be more extensive and it could be structured differently, with more sections and more topics related to plant sterols.

Author Response

REVIEWER 01

Dear Reviewer,

We appreciate you for the supportive comments and recommendations, that help us to improve our review in ways of clarity and discussion of recently skipped issues in plant sterols effectiveness. Below we provided our responses to your comments and hope they satisfy your expectations.

Comments and Suggestions for Authors

Q1: My major concern is the low quality of the Figures, making it difficult to see, and the letters are very small in Figure 1.

A1: Dear Reviewer, we agree with this claim. After careful review of illustrations, we decided to modified all figures (except Figure 4) not limiting by the Figure 1. We made the font size bigger and changed the font type to make it more contrast and attractive. Also, we modified arrows type in Figure 1 and made the starting point to indicate the action wherever appropriate. In the Figure 2 we removed excessive details and inserted a descriptive scheme of HDL-like particle with plant sterols inclusions. The last figure (Figure 5) has also been modified in line of the font size enlargement and contrasting the background. We hope, that modified figures are more readable and perceived in the current state.

Q2: I think that in general it is an interesting topic and that it is well written. Perhaps, the conclusions could be more extensive and it could be structured differently, with more sections and more topics related to plant sterols.

A2: We appreciate the Reviewer for this recommendation and added a short discussion in the “Conclusion” section, focused on individual variability in response to functional food and different bioactive compounds in the plant-based food. We suggest, that numerous factors, including nutrigenomic, epigenetic, ecological, ethnic, etc., can be responsible for such variability widely reported in sterols effect research, and, therefore, may determine ambiguity of sterols effect observed in many reports. Assumingly, the combination of deeper insights in molecular mechanisms of plants sterols (or any plant-based bioactive compound) transformation in human, and comprehensive consideration of nutrigenomic and ecological factors should be the central point in the nearest future when developing personalized dietary recommendations or preferences focused on the specific task in human health maintenance. This side of nutrition has not been touched in our review, but apparently, that is important point when talking about plant food and, particularly, plant sterols.

We also subdivided the section 4 according to the different effects of plants sterols (e.i., cardioprotective, neuromodulation, skin regeneration). Due the cardioprotective role of plant sterols was the least merited, we extended this topic (subsection 4.1 in the revised manuscript) and added several new relevant references.

Please, pay attention to the revised Table 1 since the Table has been substantially modified. We ordered all references according to the type of research (research study, clinical trial, comparative study, meta-analysis), indicated was the study conducted on human subjects or animal model; added the new column reporting the purpose of study; and revised detail in the “Observed effect” column.

Reviewer 2 Report

The authors comprehensively review the diversity of plant sterols.

I interestingly read the article. There was a little research about plant sterols.

I think this work contribute to the research about the effect of plant sterols on human health.

However, there are several concerns.

I think the references are needed in the introduction section.

The authors mentioned they reviewed the researches during the ten years. Please specify the exact study duration that the authors reviewed (e.g. studies since 2010) in the introduction section or method section.

I think to separate the study design and treatment characteristics of study subject in Table 1.

The table 1 has to contain more detailed information about each study.

The resolution of the picture is too low.

Figure 2: I could not understand the exact meaning the color of circle. What the color of circle indicate in this figure?

More detailed figure legend is needed.

Line 217: Which figure do you intend?

I think the section 4 could be improved by compartmentation according to diseases. And

Table 1 should be described in more detail.

Author Response

REVIEWER 02

Dear Reviewer,

We thank you for the valuable comments and criticism. Considering your recommendations, we could revise and improve our review and raise the issues of plant sterols efficacy and response interindividual variability. Below we provided our responses to your comments and hope they satisfy your expectations.

Comments and Suggestions for Authors

The authors comprehensively review the diversity of plant sterols. I interestingly read the article. There was a little research about plant sterols. I think this work contribute to the research about the effect of plant sterols on human health. However, there are several concerns.

  • Q1: I think the references are needed in the introduction section.
  • A1: We added appropriate references in the “Introduction” section and slightly modified this section by adding information about the most frequent explanation of controversial effects of plant sterols, possible relationship to gut microbiota and some nutrigenetic factors affecting the availability and responsiveness to plant sterols. The newly added information also has been accompanied by the relevant references.

  • Q2: The authors mentioned they reviewed the researches during the ten years. Please specify the exact study duration that the authors reviewed (e.g., studies since 2010) in the introduction section or method section.
  • A2: We agree with this claim, since the Review article must indicate the encompassed time range. Therefore, we added this information in the last sentence of the “Introduction” section as following: “[…] and the possible mechanisms of their action mentioned in different clinical trials, meta-analysis and research studies starting from the 2010 year.”

  • Q3: I think to separate the study design and treatment characteristics of study subject in Table 1.
  • A3: We modified Table 1 according to the recommendation. First, the type of study (clinical trial, research study, etc.) is now indicated in a separate specific column “Type of study”. Second, studies are ordered according to the attributed type, starting from “review”, “meta-analysis” and up to “clinical trial”, “research study” and “comparative study”. Moreover, all studies are organized according to the subjects of study, i.e., human subjects or animal model.

  • Q4: The table 1 has to contain more detailed information about each study.
  • A4: We completely agree with this recommendation. Therefore, we substantially modified the content of Table 1. Apart some modifications, mentioned in the previous reply (A3), we added the new column “Purpose of the study”. In this respect, we indicated the main aim of the study. Also, we modified information and added more details in “Observed effect” and changed details in “Characterization and study design”. The latter now contains brief information about subjects, treatment, dosage, model (if any). The “Purpose of study” now demonstrates what the main focus of research and why that was important. The “Observed effect” column now shows main achievements and results of the study exactly in the context of purpose of the study. We suggest, the revised Table 1 now contains sufficient information for the brief overview pf plant sterols application in different fields and on various subjects (human, animals, cells culture).

  • Q5: The resolution of the picture is too low.
  • A5: We have modified all figures (except Figure 3 and Figure 4) and improved the quality by changing the font type and size, contrasting them and removing excessive details. We also exported revised figures at a higher resolution of 400 dpi. We hope, that the modified figures meet the criteria of quality and become more readable. If necessary, we may provide to Editorial Board the separated file of each original image for handling.

  • Q6: Figure 2: I could not understand the exact meaning the color of circle. What the color of circle indicates in this figure? More detailed figure legend is needed.
  • A6: We are sorry for these misleading circles. In fact, circles do not include any color-coding; they are simply bullets belonging to the list of main actions of plant sterols on the central nervous system. To avoid further confusions, we made the white color uniformly for these bullets, and decreased the size of circles.

  • Q7: Line 217: Which figure do you intend?
  • A7: Thank you for this correction. That should be Figure 2; we fixed it and also checked figures indication throughout the text.

  • Q8: I think the section 4 could be improved by compartmentation according to diseases.
  • A8: We agree with the Reviewer’s opinion and divided the section number 4 in three separate subsections:

4.1 Cardioprotective property of plants sterols;

4.2 Neuroprotection and neuroimmunomodulation;

4.3 Anti-aging and skin regeneration effects.

However, since the cardioprotective effect of plant sterols merited the least attention, we extended this subsection and added a portion of relevant information regarding cholesterol-lowering effect and great variability observed in numerous publications reporting cardioprotective effect of sterols consumption.

  • Q9: Table 1 should be described in more detail.
  • A9: We added more details and essential information in Table 1 to make to more incorporated and properly ascribed in the context of plant sterol properties throughout the text.

Dear Reviewer, please, pay attention to the “Conclusion” section, because we significantly revised this section and re-wrote it in a more extensive manner. We mentioned the issue of interindividual variability reported in many studies, and raised the most relevant and frequently mentioned challenges related to evolution of plant sterols effect. Although, our review does not touch the issue of sterols availability, delivery and nutrigenomic factor, we assume that this problems merit attention as they are part of plant sterol effectiveness and responsiveness to sterol-enriched food.

Round 2

Reviewer 2 Report

Thank you for appropriate revision.